# Noise Parameter Estimation Two-Stage Network for Single Infrared Dim Small Target Image Destriping

**Teliang Wang, Qian Yin, Fanzhi Cao, Miao Li, Zaiping Lin * and Wei An**

The College of Electronic Science and Technology, National University of Defense Technology, Changsha 410073, China
* Correspondence: zaipinglin08@nudt.edu.cn

**Abstract:** The existing nonuniformity correction methods generally have the defects of image blur, artifacts, image over-smoothing, and nonuniform residuals. It is difficult for these methods to meet the requirements of image enhancement in various complex application scenarios. In particular, when these methods are applied to dim small target images, they may remove dim small targets as noise points due to the image over-smoothing. This paper draws on the idea of a residual network and proposes a two-stage learning network based on the imaging mechanism of an infrared line-scan system. We adopt a multi-scale feature extraction unit and design a gain correction sub-network and an offset correction sub-network, respectively. Then, we pre-train the two sub-networks independently. Finally, we cascade the two sub-networks into a two-stage network and train it. The experimental results show that the PSNR gain of our method can reach more than 15 dB, and it can achieve excellent performance in different backgrounds and different intensities of nonuniform noise. Moreover, our method can avoid losing texture details or dim small targets after effectively removing nonuniform noise.

**Keywords:** nonuniformity correction; deep learning; dim small target; image over-smoothing; infrared line-scan image

## 1. Introduction

With the development of infrared remote sensing detection systems, infrared line-scan detectors have been widely used in military and civilian fields such as battlefield surveying, maritime surveillance, and urban traffic monitoring. However, restricted by factors such as the level of manufacturing process, there is nonuniform noise in the response between the detection elements of the line-scan sensor, resulting in a certain strip effect in the original infrared image obtained. This requires that we need to first perform nonuniformity correction on the image before performing target detection [1]. In addition, the infrared remote sensing detection system not only has a long observation distance, but also is often interfered with by complex background clutter and noise. Therefore, the targets often show the characteristics of dim small targets such as low signal-to-noise ratio and lack of effective structural information. According to the definition of the Society of Photo-Optical Instrumentation Engineers (SPIE), the target with a local signal-to-noise ratio <5 dB and a pixel size $\leq 9 \times 9$ is regarded as a weak target [2]. These characteristics of dim small targets make them very easy to be regarded as noise points, which requires us to pay special attention to the problem of image over-smoothing when removing strip noise.

For the fixed pattern noise (FPN) generated by infrared imaging systems, researchers have proposed two kinds of nonuniformity correction methods, which are calibration-based and scene-based [3].

The calibration-based correction method utilizes the two-point or multi-point response between detection elements of the sensor to the black body radiation source and calculates the correction parameters through mathematical fitting. The advantage of this kind of

method is that the algorithm is simple, but its correction performance is affected by factors such as ambient temperature and integration time changes. Therefore, the calibration-based correction method needs to be performed periodically, and the detection system needs to suspend the mission work while observing the blackbody radiation source for calibration [4,5].

The scene-based correction methods can be divided into the multi-frame method and single-frame method, which correct the nonuniformity of the image according to the correlation of the scene. Since these kinds of methods do not need to observe the calibration source, they has the advantage of not affecting the normal operation of the infrared system.

The classic multi-frame methods mainly include Neural Network [6], Constant Statistics [7], Kalman Filtering [8], and Inter-frame Registration Algorithm [9]. However, the multi-frame methods produce artifacts when the scene motion changes suddenly. In addition, due to the complexity of the algorithm, the real-time performance of such methods in practical applications is often not ideal. Although some recent multi-frame methods such as the Adaptive Deghosting Method in Neural Network [10], Temporal–Spatial Nonlinear Filtering [11], parameter estimation [12], and Feature Pattern Matching [13] have made some improvements to these shortcomings, in general, the single-frame method is still the main research direction at present.

Single-frame methods mainly include Midway Histogram Equalization (MHE) [14], 1-D Guided Filtering (1D-GF) [15], Sparse Unidirectional Hybrid Total Variation [16], Enhanced Low-rank Prior and Total Variation Regulation [17], Wavelet Decomposition and Total Variation-Guided Filtering [18], Least Squares and Gradient Domain-Guided Filtering [19], and so on. In addition, there are some methods that focus on over-smoothing. For example, Mingxuan Li et al. proposed an adaptive edge-preserving operator (AEPO) based on edge contrast to prevent the loss of edge details [20]. In another paper, the Multi-Scale Wavelet Transform adopted by Mingxuan Li et al., can also protect non-stripe information in infrared images [21]. The Fuzzy Matrix and Statistical Filtering method proposed by Sichong Huang et al. also has a certain ability to retain detailed information [22].

In recent years, with the success of deep learning methods in various fields such as image classification [23], target detection [24], face recognition [25], and even short-term power load forecasting [26], more and more researchers are also applying deep learning to the field of image nonuniformity correction. Currently, open-source deep learning nonuniformity correction methods include SNRCNN [27], ICSRN [28], DLS-NUC [29], and SNRWDNN [30]. In addition, there are Deep Multi-scale Residual Network [31], Cascade Residual Attention CNN [32], Deep Multi-scale Dense Connection CNN [33], and so on. Significantly, Timing Li et al. proposed that the use of long connections in deep networks can solve the problem of image information loss caused by transposed convolution [34], and Sifan Zhang et al. proposed a regularization method based on features extracted by wavelet to help restore image details [35].

However, although some existing methods have achieved certain results in preserving image details, they have not fundamentally solved the problem of image over-smoothing. In this regard, based on the nonuniform noise model of line-scan detectors, we transform the problem of nonuniformity correction into the problem of estimating noise parameters. We designed a two-stage fully convolutional network including a gain correction sub-network and an offset correction sub-network. At the same time, we produced datasets to pre-train and train the network. In general, our contributions are as follows:

1. A destriping method based on a noise parameter estimation two-stage network is proposed, which can adapt to the input image size and effectively correct the real nonuniformity infrared image;
2. According to the nonuniformity response model of the line-scan detector, a deep learning dataset for strip noise parameter estimation and image reconstruction is produced;
3. A multi-scale feature extraction unit is designed to use image information more effectively, and the proposed network has excellent generalization to different intensities of nonuniform noise and different backgrounds;

4. The noise parameter estimation mechanism in our network can fundamentally solve the problem that texture details and dim small targets may be removed due to image over-smoothing.

## 2. Methods

In this paper, two deep learning sub-networks for estimating the gain correction coefficient and estimating the offset correction factor are designed, respectively. After pre-training the sub-networks, we cascade the two sub-networks through a multiplication structure and an addition structure as a two-stage network. Finally, we train the two-stage network, and the resulting network model can perform nonuniformity correction on real infrared images.

### 2.1. Nonuniformity Response Model and Datasets

Our two-stage deep learning network model requires a large amount of data for training; however, due to less research on deep learning methods for nonuniformity correction, no ready-made datasets are available. We downloaded the MS-COCO datasets [36], converted the images in the datasets into grayscale images as clean infrared images, and then added nonuniform noise to these clean infrared images according to the nonuniformity response model of the infrared line-scan detector to produce our datasets.

The nonuniformity response of an infrared line-scan detector can be expressed as follows:

$$y_{ij} = g_i x_{ij} + o_i, \tag{1}$$

where $x_{ij}$ and $y_{ij}$ represent the actual response and observed value of the detector unit $i$ scanned to the row $j$, respectively; $g_i$ and $o_i$ represent the gain and offset of the detector unit $i$, respectively.

The image nonuniformity correction is essentially to remove the nonuniform noise generated by each detector unit shown in Equation (1) from the observed signal $y_{ij}$, so as to obtain the estimated value $\hat{x}_{ij}$ of the actual response $x_{ij}$. The process can be expressed as follows:

$$\hat{x}_{ij} = G_i y_{ij} + O_i, \tag{2}$$

where $G_i$ and $O_i$ are the gain correction coefficient and the offset correction factor, respectively:

$$G_i = 1/g_i, \tag{3}$$

$$O_i = -o_i/g_i, \tag{4}$$

Therefore, the problem of image nonuniformity correction is the problem of estimating the gain correction coefficient $G_i$ and the offset correction factor $O_i$.

Based on the above nonuniformity response model, we produced our datasets. First, we cropped the image of size $128 \times 128$ from the image in the MS-COCO dataset. Second, we converted the cropped image to grayscale image, and we backed up the grayscale image as "Ori". Third, we added noise to the grayscale image according to Equation (1) and save the noised image as "Nuf". At the same time, we substituted the noise factors $g_i$ and $o_i$ into Equation (3) and Equation (4), respectively, for calculation to obtain correction factors $G_i$ and $O_i$. Finally, we obtained the mat data of {"Nuf","Ori", "G", "O"}, where "Nuf" are grayscale image of size $128 \times 128$ with nonuniform noise added, "Ori" are the corresponding un-noised original grayscale image of size $128 \times 128$, "G" are the corresponding nonuniformity gain correction coefficients of size $1 \times 128$, and "O" are the corresponding nonuniformity offset correction factors of size $1 \times 128$. The specific operation of producing the datasets in this paper is shown in Figure 1.

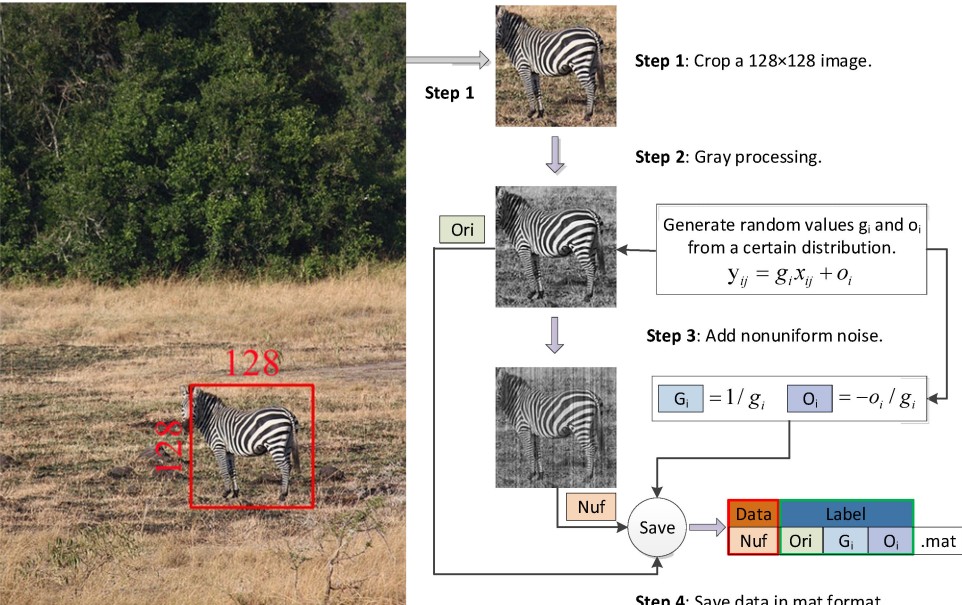

Step 1: Crop a 128×128 image.

Step 2: Gray processing.

Generate random values $g_i$ and $o_i$ from a certain distribution.
$$\mathbf{y}_{ij} = g_i x_{ij} + o_i$$

Step 3: Add nonuniform noise.

$G_i = 1/g_i$　$O_i = -o_i/g_i$

Step 4: Save data in mat format.

**Figure 1.** The specific operation of producing the datasets.

It should be noted that in the step 3, the multiplicative noise $g_i$ of the same image obeys the uniform distribution of $\left(1 - \sigma_g, 1 + \sigma_g\right)$, the additive noise $o_i$ of the same image follows a Gaussian distribution with a mean of 0 and a standard deviation of $\sigma_o$, and each image corresponds to a different random standard deviation, $\sigma_g \in [0, 0.2]$, $\sigma_o \in [0, 32]$.

As above, we generated 500,000 nonuniformity images, of which 200,000 images were used to pre-train our two sub-networks, and the other 300,000 images were used to train the two-stage network with the two sub-networks cascaded.

## 2.2. Network Design

Our two-stage network mainly consists of two parts, including a gain correction sub-network and an offset correction sub-network. Our network adopts the design principle of fully convolutional network (FCN), which can adapt to the size of the input images [37]. In our network, the convolutional layer is a multi-scale feature extraction unit, as shown in Figure 2.

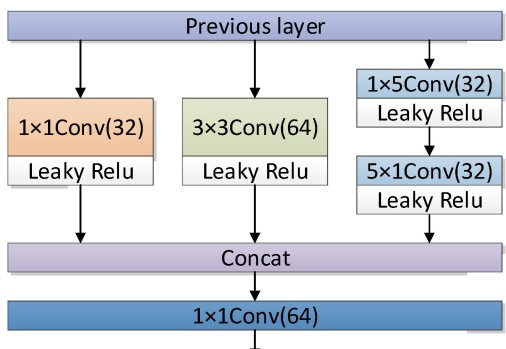

**Figure 2.** A multi-scale feature extraction unit.

As can be seen from Figure 2, we cascade 32 asymmetric convolution kernels of $1 \times 5$ and 32 asymmetric convolution kernels of $5 \times 1$ and connect the cascade with 32 convolution kernels of $1 \times 1$ and 64 convolution kernels of $3 \times 3$ in parallel. In addition, the cascade of asymmetric convolution kernel of $1 \times 5$ and asymmetric convolution kernel of $5 \times 1$ is equivalent to a $5 \times 5$ convolution kernel in the perceived field of view [38]. Moreover, each convolution operation is followed by a Leaky ReLU activation function [39] to introduce nonlinearity. Then, we concatenate the multi-scale features obtained by

convolution in the channel direction. At last, we adopt a convolution kernel of $1 \times 1$ to fuse the features across channels and reduce the dimension. By adopting multi-scale feature extraction units, we can not only increase the adaptability to targets of different scales, but also improve the network expression ability without increasing the network depth to avoid the phenomenon of gradient dispersion.

Shown in the Figure 3 is the gain correction sub-network structure. We use 8 convolution layers with multiple feature extraction units, and then use a convolution kernel of $1 \times 1$ to reduce the output to a single channel, so as to obtain an output with the same size as the input image. At last, we use the column mean of this output as an estimate of the gain correction coefficient.

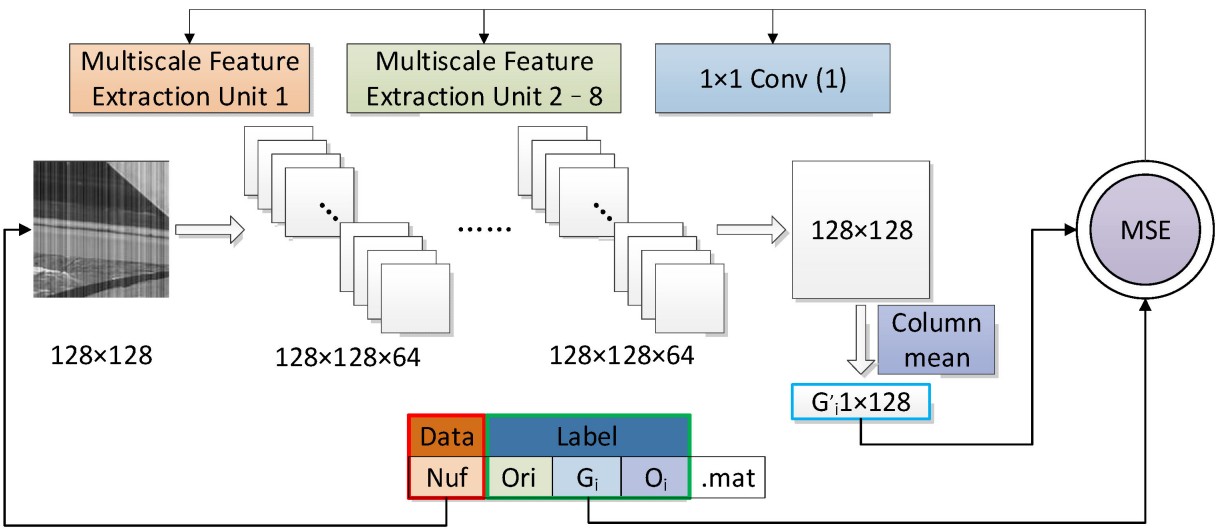

**Figure 3.** The gain correction sub-network structure.

In this sub-network, we used the nonuniformity infrared image "Nuf" as input, the saved gain correction coefficient "$G_i$" as the true value label, and the mean square error as the loss function for pre-training, where the loss function of the gain correction sub-network can be expressed as:

$$L_{G-MSE} = (1/W)\sum_{i=1}^{W}\left(G_i' - G_i\right)^2,\tag{5}$$

where $W$ is the width of the image, $G_i'$ is the estimated value of the gain correction coefficient output by the gain correction sub-network, and $G_i$ is the saved true value of the gain correction coefficient.

Shown in the Figure 4 is the offset correction sub-network structure. In this sub-network, we use 15 convolution layers with multi-feature extraction units and a convolution kernel of $1 \times 1$, and finally use the column mean of the output as the estimate of the offset correction factor.

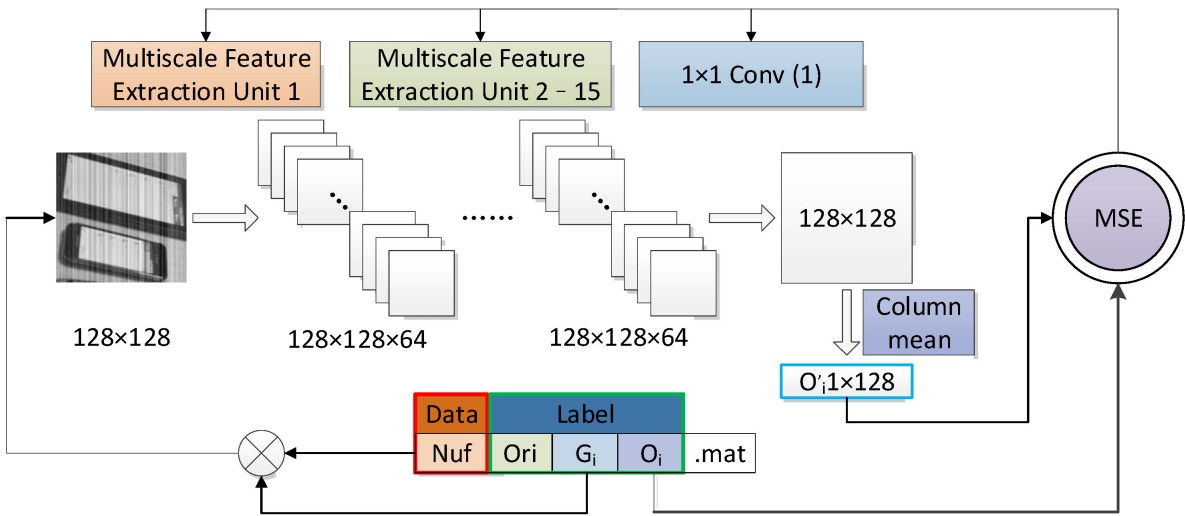

**Figure 4.** The offset correction sub-network structure.

Since the offset correction sub-network and the gain correction sub-network are pretrained independently, both sub-networks can use the same nonuniformity infrared image datasets. Different from the gain correction sub-network, the offset correction sub-network uses the image obtained by multiplying the saved nonuniformity infrared image "Nuf" and the gain correction coefficient "$G_i$" as input and uses the saved offset correction factor "$O_i$" as the true value label. In this sub-network, we also use the mean square error as the loss function for pre-training, as shown in the formula:

$$L_{O-MSE} = (1/W)\sum_{i=1}^{W}\left(O_i' - O_i\right)^2,  \tag{6}$$

where $W$ is the width of the image, $O_i'$ is the estimated value of the offset correction coefficient output by the offset correction sub-network, and $O_i$ is the saved true value of the offset correction coefficient.

Shown in Figure 5 is the two-stage deep learning network structure of gain and offset correction. In our network, since the gain correction coefficient and offset correction factor of the line-scan detector are estimated by column, and the "small" feature of the dim small target makes it impossible to have a decisive influence on the estimation of the column, the problem that may lead to image over-smoothing and remove the dim small targets when reconstructing the images pixel by pixel based on image correlation in traditional methods can be avoided.

It is important to emphasize that the training set of the cascaded two-stage network cannot be the same as the training set of the two sub-networks. When training the cascaded two-stage network, we use the nonuniformity infrared image "Nuf" as input, use the saved clean infrared image "Ori" as the true value label, and use the mean square error as the loss function, as shown in the formula:

$$L_{I-MSE} = [1/(W{\cdot}H)]\sum_{i=1}^{W}\sum_{j=1}^{H}\left(I_{ij}' - I_{ij}\right)^2,  \tag{7}$$

where $W$ is the width of the image, $H$ is the height of the image, $I_{ij}'$ is the reconstructed image output by the network model, $I_{ij}$ is the saved clean image, and $i$ and $j$ are the pixel coordinates of the image.

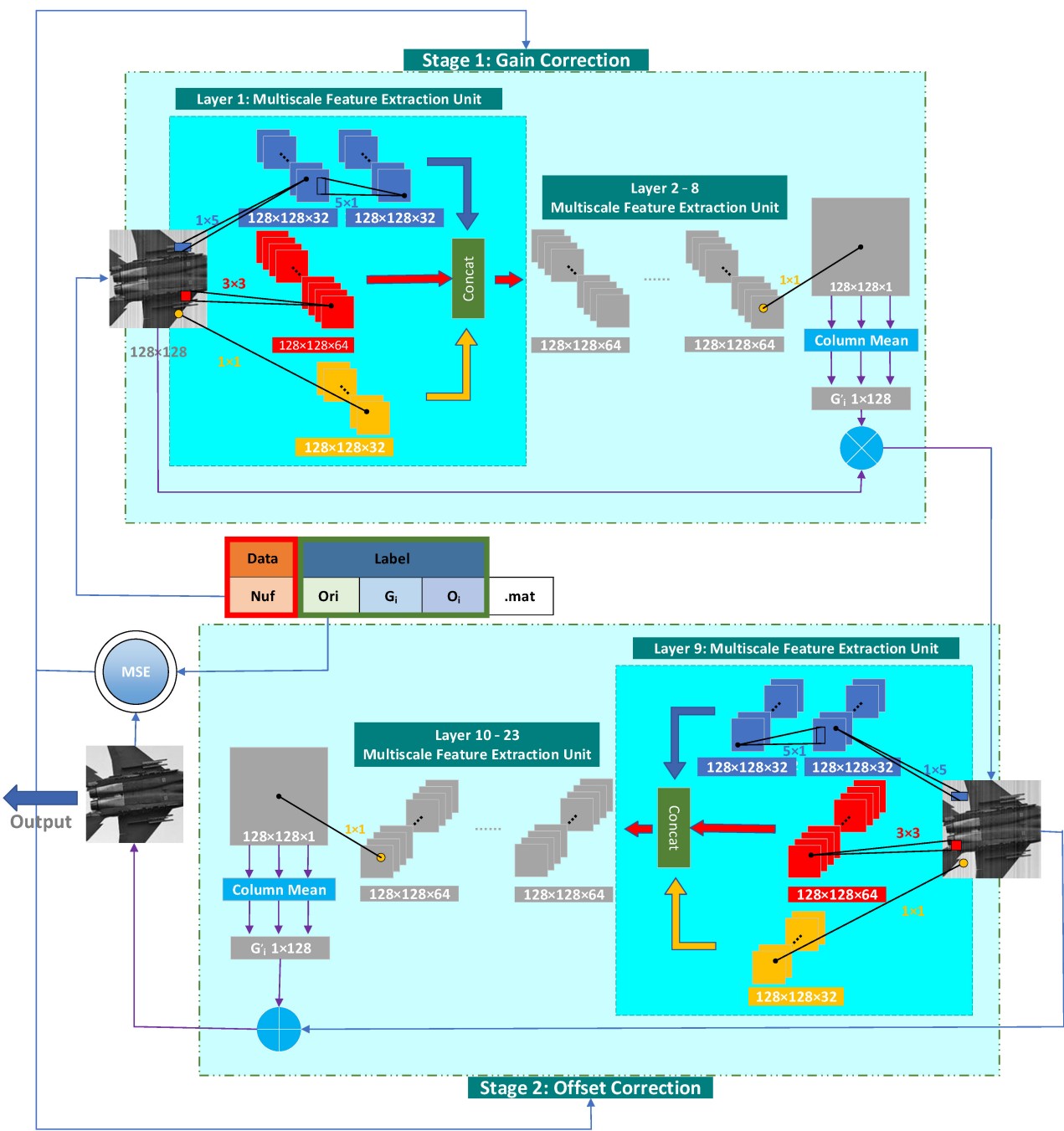

**Figure 5.** The two-stage deep learning network structure of gain and offset correction.

### 3. Results

In experiments, to verify the effectiveness of our method, we selected the most commonly used traditional methods for comparison, including MHE [14] and 1D-GF [15]. At the same time, we also compare our method with already open-source deep learning methods, including SNRCNN [27], ICSRN [28], DLS-NUC [29], and SNRWDNN [30].

#### 3.1. Network Model Training

We used TensorFlow software (San Francisco, CA, USA) to build the network model and used the adaptive momentum optimizer (AdamOptimizer) in TensorFlow to pre-train and train the network model on a single GPU NVIDIA Tesla V100S-PCIe-32GB (Santa Clara, CA, USA).

In the pre-training of the gain correction sub-network and the offset correction sub-network, we randomly initialized the parameters: the batch number was set to 16, the

initial learning rate was set to 0.001, the learning rate was reduced to 0.0001 after 25 epochs, and the total number of training epochs was set to 50. In the training of the cascaded two-stage network, we initialized with the parameters obtained in the pre-training of the two sub-networks: the number of batches was set to 16, the initial learning rate was set to 0.0001, the learning rate was reduced to 0.00001 after 25 epochs, and the total number of training epochs was set to 50.

### 3.2. Quality Evaluation Metrics

In the experiments on simulated nonuniformity infrared dim small target images, we mainly compare the correction performance of each method for different intensity nonuniformity, and the influence of these methods on the dim small targets in nonuniformity infrared images. In the simulation experiments, we conducted a comparative analysis through five objective evaluation metrics, including root mean square error (RMSE) [40], peak signal-to-noise ratio (PSNR) [41], structural similarity (SSIM) [42], image roughness (IR) [43], and signal-to-clutter ratio (SCR) [44,45].

RMSE is the root mean square error of the clean reference image and the de-noised image, which is the error of image reconstruction. The smaller the RMSE, the better the performance of image reconstruction, which is defined as:

$$RMSE = \sqrt{[1/(W \cdot H)] \sum_{i=1}^{W} \sum_{j=1}^{H} (e_{ij} - r_{ij})^2}, \tag{8}$$

where $W$ is the width of the image, $H$ is the height of the image, $e_{ij}$ is the image to be evaluated output by the network model, $r_{ij}$ is the clean image for reference, and $i$ and $j$ are the pixel coordinates of the image.

PSNR is one of the most commonly used evaluation metrics for image reconstruction quality, which is calculated from RMSE. The larger the PSNR, the better the performance of image reconstruction, which is defined as:

$$PSNR = 20 \times \lg(255/RMSE), \tag{9}$$

Structural similarity is a metric to evaluate the similarity of the clean reference image and the de-noised image based on the characteristics of the human visual system, and mainly considers the three key features of image luminance, contrast, and structure. The value range of the structural similarity is (0, 1); the larger the value, the more similar the two images are, and the better the image reconstruction performance is. The definition is as follows:

$$SSIM(r, e) = \frac{(2\mu_r\mu_e + c_1)(2\sigma_{re} + c_2)}{(\mu_r^2 + \mu_e^2 + c_1)(\sigma_r^2 + \sigma_e^2 + c_2)}, \tag{10}$$

where $r$ and $e$ are the image for reference and the image to be evaluated, respectively; $\mu_r$ and $\mu_e$ are the mean of $r$ and $e$, respectively; $\sigma_r^2$ and $\sigma_e^2$ are the variance of $r$ and $e$, respectively; $\sigma_{re}$ is the covariance of $r$ and $e$. Constraints $c_1 = (k_1 L)^2$, $c_2 = (k_2 L)^2$, $L = 255$ (for 8-bit grayscale images) are the dynamic range of image pixel values, and the default values for $k_1$ and $k_2$ are 0.01 and 0.03, respectively.

Image roughness is an important metric to measure image sharpness. The smaller the image roughness, the better the image quality, which is defined as:

$$IR = \frac{\|h * e\|_1 + \|h^T * e\|_1}{\|e\|_1}, \tag{11}$$

where $e$ is the reconstructed image to be evaluated, $h = [1, -1]$ is the horizontal mask, $h^T$ as the transpose of $h$ is the vertical mask, the asterisk denotes discrete convolution, $\| \|_1$ denotes the $L_1$ norm, $\|e\|_1$ is the sum of all pixel values of the image, and $\|h * e\|_1$ and

$\|h^T * e\|_1$ are the sum of differences between pixels in the horizontal and vertical directions of the reconstructed image, respectively.

The signal-to-clutter ratio is an important metric to measure the difficulty of detecting dim small targets. The higher the signal-to-clutter ratio of dim small targets, the easier it is to be detected. Its definition is as follows:

$$SCR = \frac{|\mu_t - \mu_b|}{\sigma_b}, \tag{12}$$

where $\mu_t$ is the average pixel value of the target, $\mu_b$ is the average pixel value of the target neighborhood background, and $\sigma_b$ is the standard deviation of the pixel value of the target neighborhood background. Figure 6 shows the small target and its neighborhood background, in which the height $h$ and width $w$ of the small target are not greater than 5, and we set the width $d$ of the neighborhood background as 5.

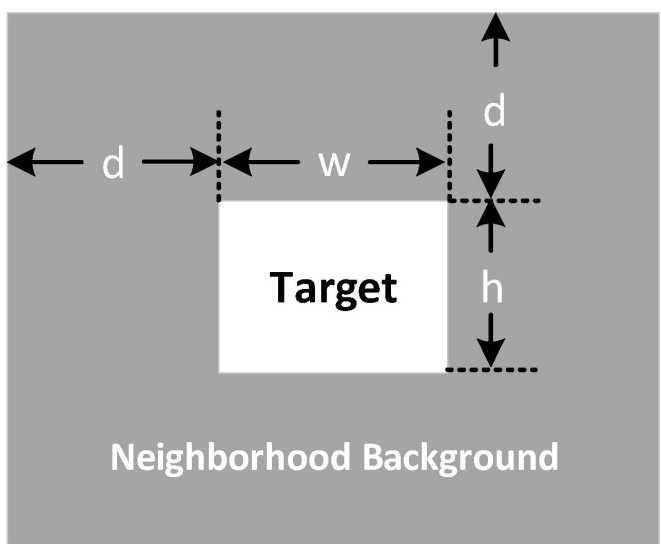

**Figure 6.** Small target and its neighborhood background.

In the experiments on real nonuniformity infrared images, we mainly verified the correction performance of each method on real images and compared the effects of each method on texture details. Since there is no corresponding clean infrared image for the real nonuniformity infrared image, we can only use non-reference evaluation metrics. In the field of image processing, compared with the reference index, the non-reference index generally has certain defects. In addition to the above *IR* metrics, in order to comprehensively evaluate the denoising performance and the ability to preserve image details of each method, we also adopted the inverse coefficient of variation (ICV) [46,47] and the mean relative deviation (MRD)) [48]. However, all of these metrics are affected by factors other than algorithm performance. Therefore, we performed edge detection [49] on the images processed by each method to help us compare the impact of each method on image texture details more objectively.

The inverse coefficient of variation (ICV) is an indicator for evaluating the smoothness of an image homogeneous region, which can measure the destriping performance of each method. It is defined as:

$$ICV = \frac{\mu_h}{\sigma_h}, \tag{13}$$

where $\mu_h$ and $\sigma_h$ are the mean and standard deviation of pixel values in a homogeneous region, respectively. In general, the larger the ICV, the smoother the homogeneous region of the image, and the better the destriping performance.

In contrast to ICV, the mean relative deviation (MRD) is an indicator for evaluating the relative distortion of a sharp region, which is defined as:

$$MRD = [1/(W \cdot H)]\frac{\sum\limits_{i=1}^{W}\sum\limits_{j=1}^{H}\left|s'_{ij} - s_{ij}\right|}{s_{ij}}, \tag{14}$$

where $s'_{ij}$ and $s_{ij}$ are the pixel pair before and after destriping in a sharp region. In general, the smaller the MRD, the less distortion in the sharp region of the image, and the better the ability to retain details.

However, although ICV and MRD are more targeted than IR metrics, these two metrics also have certain limitations due to the need for us to manually delineate the homogeneous and sharp region of the image.

Edge detection is to highlight the structural information by finding the points with obvious brightness changes in the image. In this paper, we use the Sobel operator for edge detection. We convolve the image to be detected with the Sobel operator to obtain the gradient values of each pixel in the horizontal and vertical directions, and then use the gradient of each pixel as the pixel value of the edge detection image, as shown in the formula:

$$\left.\begin{array}{l}G_x(x,y) = e(x,y) \otimes g_x \\ G_y(x,y) = e(x,y) \otimes g_y\end{array}\right\} \Rightarrow EI(x,y) = \sqrt{G_x^2(x,y) + G_y^2(x,y)}, \tag{15}$$

where the Sobel operator is as follows:

$$g_x = \begin{bmatrix} -1 & 0 & 1 \\ -2 & 0 & 2 \\ -1 & 0 & 1 \end{bmatrix}, \ g_y = \begin{bmatrix} -1 & -2 & -1 \\ 0 & 0 & 0 \\ 1 & 2 & 1 \end{bmatrix}, \tag{16}$$

### 3.3. Method Comparison on Simulated Data with Different Intensities of Nonuniformity

We selected one real, clean infrared small target image from the SIRST dataset [50]. As shown in Figure 7, the dim small target is located in the middle of the red box we marked.

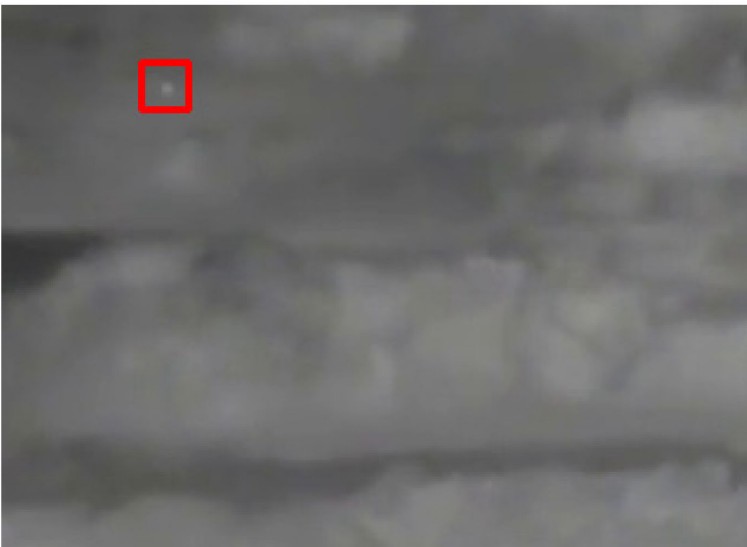

**Figure 7.** A real, clean infrared small target image.

We added different intensities of nonuniform noise to this image. As a low-intensity nonuniformity, its multiplicative noise $g_i$ obeys a uniform distribution of $(1 - 0.05, 1 + 0.05)$, and its additive noise $o_i$ obeys a Gaussian distribution with a mean of 0 and a standard deviation of 5. As a medium-intensity nonuniformity, its multiplicative noise $g_i$ obeys

a uniform distribution of $(1 - 0.10, 1 + 0.10)$, and its additive noise $o_i$ obeys a Gaussian distribution with a mean of 0 and a standard deviation of 15. As a high-intensity nonuniformity, its multiplicative noise $g_i$ obeys a uniform distribution of $(1 - 0.15, 1 + 0.15)$, and its additive noise $o_i$ obeys a Gaussian distribution with a mean of 0 and a standard deviation of 25.

The results of correction methods for different intensities of nonuniformity infrared dim small target images are shown as Figures 8–10, respectively. Shown in Figure 8 are the correction results of different methods on a low-intensity nonuniformity infrared dim small target image.

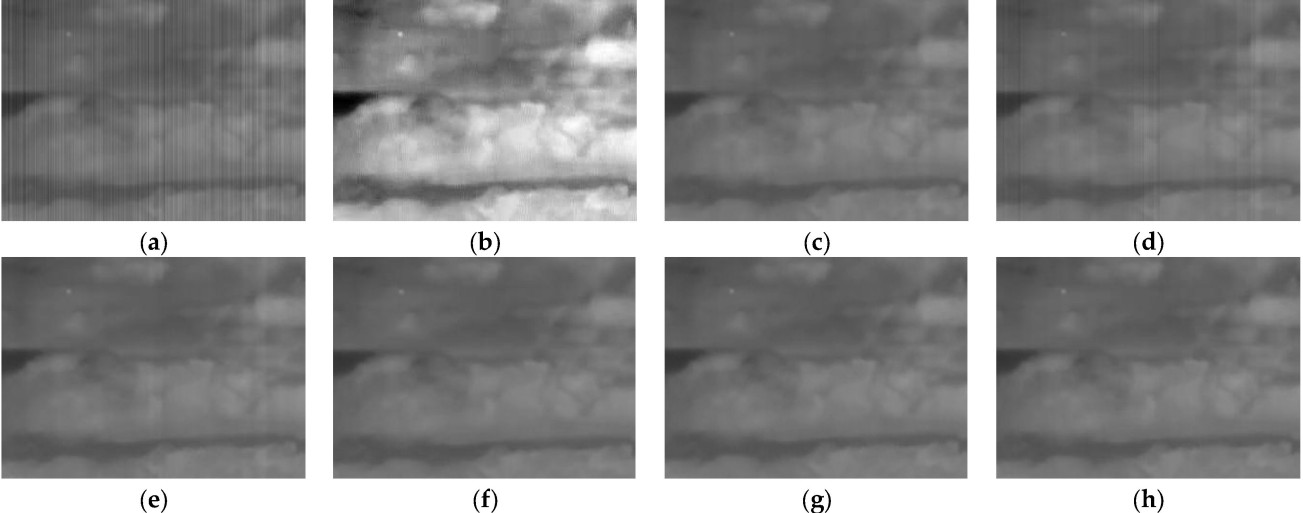

**Figure 8.** Comparison of methods using low-intensity nonuniformity infrared dim small target image: (**a**) infrared dim small target image with low-intensity nonuniform noise added; (**b**) MHE; (**c**) 1D-GF; (**d**) SNRCNN; (**e**) ICSRN; (**f**) DLS-NUC; (**g**) SNRWDNN; (**h**) ours.

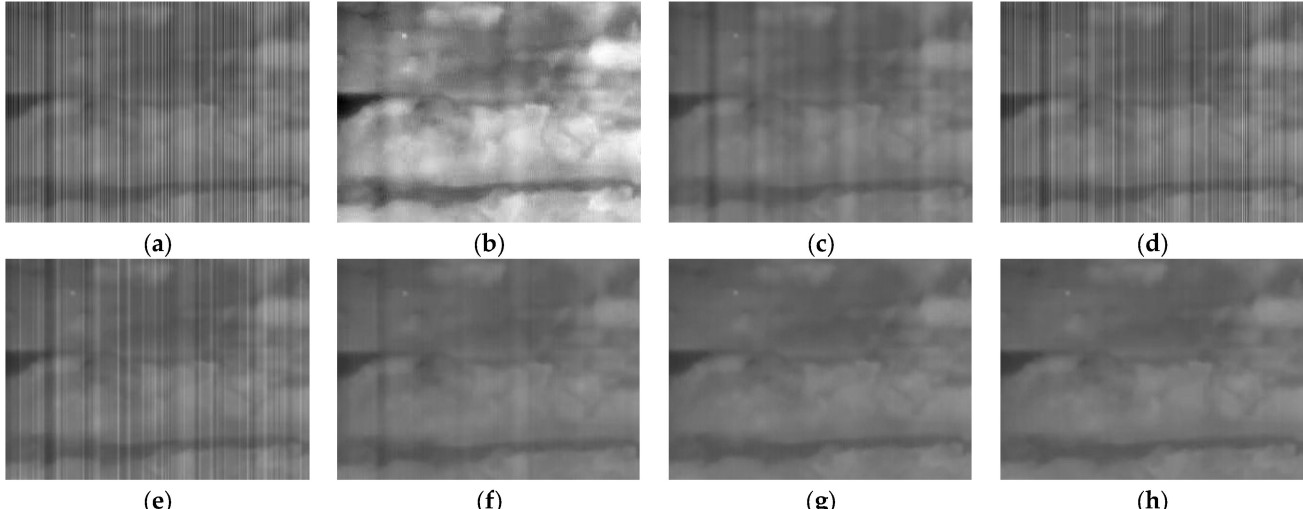

**Figure 9.** Comparison of methods using medium-intensity nonuniformity infrared dim small target images: (**a**) infrared dim small target image with medium-intensity nonuniform noise added; (**b**) MHE; (**c**) 1D-GF; (**d**) SNRCNN; (**e**) ICSRN; (**f**) DLS-NUC; (**g**) SNRWDNN; (**h**) ours.

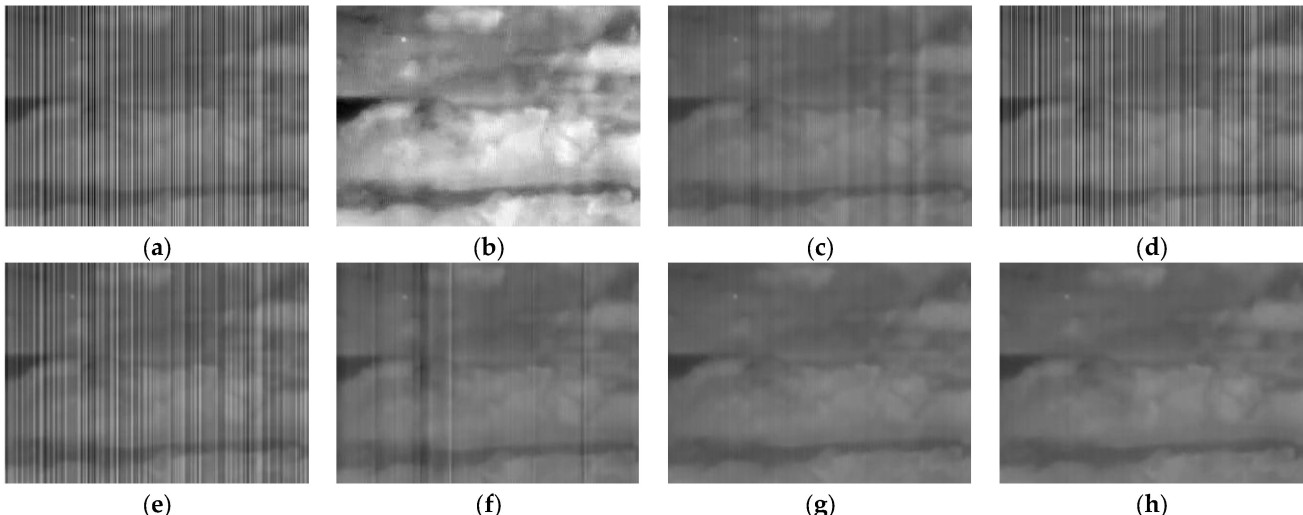

**Figure 10.** Comparison of methods using high-intensity nonuniformity infrared dim small target images: (**a**) infrared dim small target image with high-intensity nonuniform noise added; (**b**) MHE; (**c**) 1D-GF; (**d**) SNRCNN; (**e**) ICSRN; (**f**) DLS-NUC; (**g**) SNRWDNN; (**h**) ours.

Shown in Figure 9 are the correction results of different methods on a medium-intensity nonuniformity infrared dim small target image.

Shown in Figure 10 are the correction results of different methods on a high-intensity nonuniformity infrared dim small target image.

According to the evaluation metrics described in Section 3.2, we compare the correction performance of each method for different intensities of nonuniform noise, as shown in Table 1.

In order to more intuitively compare the generalization performance of each method for different intensities of nonuniformity, we created a line graph of each method regarding PSNR gain, as shown in Figure 11.

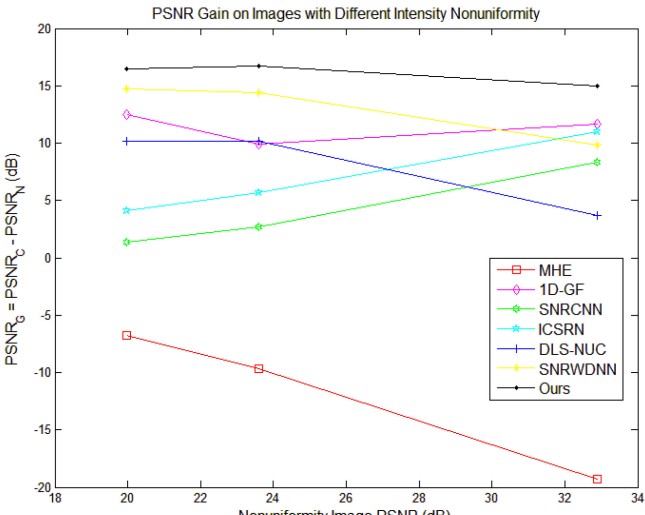

**Figure 11.** The PSNR gain of each method on images with different intensities of nonuniformity. The greater the PSNR along the abscissa direction, the lower the intensity of image nonuniformity.

### 3.4. Method Comparison on Simulated Data with Different Backgrounds

We selected 10 real, clean infrared dim small target images from the SIRST dataset [50], measured the background complexity by the pixel value variance $B_{var}$ (dynamic range of 0–255), and named these images in increasing complexity as Test-1 to Test-10. We then added equal-intensity nonuniform noise to these images. The multiplicative noise $g_i$ obeys

a uniform distribution of $(1 - 0.12, 1 + 0.12)$, and its additive noise $o_i$ obeys a Gaussian distribution with a mean of 0 and a standard deviation of 12. Figures 12–14 are the three most representative experimental results we selected.

**Table 1.** Evaluation metric results of the methods on simulated data with different intensities of nonuniform noise.

| Experiment | Methods | RMSE | PSNR (dB) | SSIM | IR | SCR |
|---|---|---|---|---|---|---|
| | Noise Image | 5.77 | 32.9012 | 0.7309 | 0.0706 | 2.5708 |
| Low-Intensity Nonuniformity | MHE [14] | 53.49 | 13.5652 | 0.7712 | 0.0381 | 3.1419 |
| | 1D-GF [15] | 1.51 | 44.5529 | 0.9965 | 0.0157 | 3.4791 |
| | SNRCNN [27] | 2.21 | 41.2492 | 0.9741 | 0.0205 | 3.0936 |
| | ICSRN [28] | 1.62 | 43.9166 | 0.9937 | 0.0143 | 3.6239 |
| | DLS-NUC [29] | 3.77 | 36.6104 | 0.9643 | 0.0145 | 3.5481 |
| | SNRWDNN [30] | 1.87 | 42.7037 | 0.9906 | 0.0157 | 3.6770 |
| | Ours | 1.03 | 47.8425 | 0.9993 | 0.0150 | 3.7283 |
| | Noise Image | 16.84 | 23.6059 | 0.2904 | 0.1885 | 1.323 |
| Medium-Intensity Nonuniformity | MHE [14] | 51.04 | 13.9724 | 0.7529 | 0.0424 | 3.3484 |
| | 1D-GF [15] | 5.40 | 33.4841 | 0.9623 | 0.0217 | 3.4443 |
| | SNRCNN [27] | 12.28 | 26.3438 | 0.4895 | 0.1183 | 1.9014 |
| | ICSRN [28] | 8.77 | 29.2671 | 0.7530 | 0.0451 | 3.2835 |
| | DLS-NUC [29] | 5.21 | 33.8014 | 0.9490 | 0.0183 | 3.4608 |
| | SNRWDNN [30] | 3.21 | 38.0057 | 0.9864 | 0.0168 | 3.8070 |
| | Ours | 2.45 | 40.3645 | 0.9978 | 0.0156 | 3.7163 |
| | Noise Image | 25.62 | 19.96 | 0.1559 | 0.2914 | 0.7753 |
| High-Intensity Nonuniformity | MHE [14] | 56.02 | 13.16 | 0.7218 | 0.0444 | 3.4024 |
| | 1D-GF [15] | 6.06 | 32.48 | 0.9137 | 0.0334 | 3.4181 |
| | SNRCNN [27] | 21.78 | 21.37 | 0.2382 | 0.2273 | 0.8014 |
| | ICSRN [28] | 16.01 | 24.05 | 0.4009 | 0.1155 | 1.1388 |
| | DLS-NUC [29] | 7.93 | 30.14 | 0.8922 | 0.0272 | 2.7439 |
| | SNRWDNN [30] | 4.72 | 34.66 | 0.9796 | 0.0191 | 3.8752 |
| | Ours | 3.83 | 36.46 | 0.9953 | 0.0170 | 3.5277 |

Figure 12 is the nonuniformity correction results of Test-1 ($B_{\mathrm{var}} = 19$).
Figure 13 is the nonuniformity correction results of Test-6 ($B_{\mathrm{var}} = 663$).
Figure 14 is the nonuniformity correction results of Test-10 ($B_{\mathrm{var}} = 2880$).

At the same time, we use the objective metrics described in Section 3.2 to quantitatively evaluate the experimental results, as shown in Table 2.

In order to compare the performance of each method on PSNR more intuitively, we created a line graph, as shown in Figure 15.

In order to verify the effectiveness of our method in solving the problem of image over-smoothing, we used the clean infrared image roughness as the abscissa reference and created a line graph for each method performance on the IR, as shown in Figure 16.

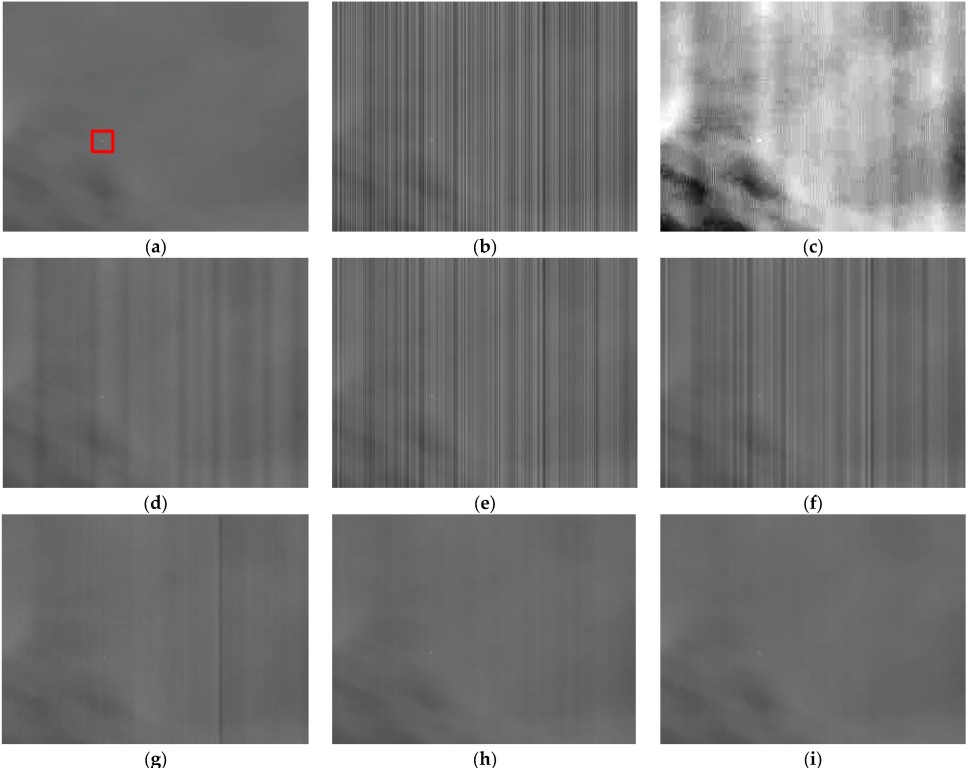

**Figure 12.** The nonuniformity correction results of Test-1: (**a**) the real, clean infrared dim small target image ($B_{var} = 19$), the dim small target is located in the middle of the red box we manually marked; (**b**) infrared dim small target image with nonuniform noise added; (**c**) MHE; (**d**) 1D-GF; (**e**) SNRCNN; (**f**) ICSRN; (**g**) DLS-NUC; (**h**) SNRWDNN; (**i**) ours.

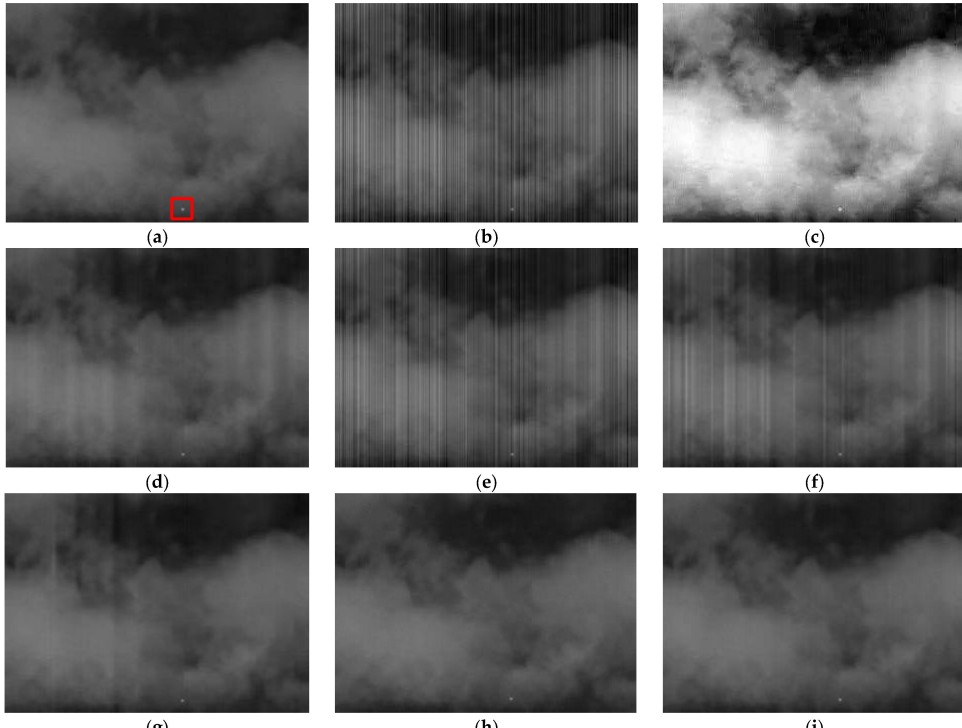

**Figure 13.** The nonuniformity correction results of Test-6: (**a**) the real, clean infrared dim small target image ($B_{var} = 663$), the dim small target is located in the middle of the red box we manually marked; (**b**) infrared dim small target image with nonuniform noise added; (**c**) MHE; (**d**) 1D-GF; (**e**) SNRCNN; (**f**) ICSRN; (**g**) DLS-NUC; (**h**) SNRWDNN; (**i**) ours.

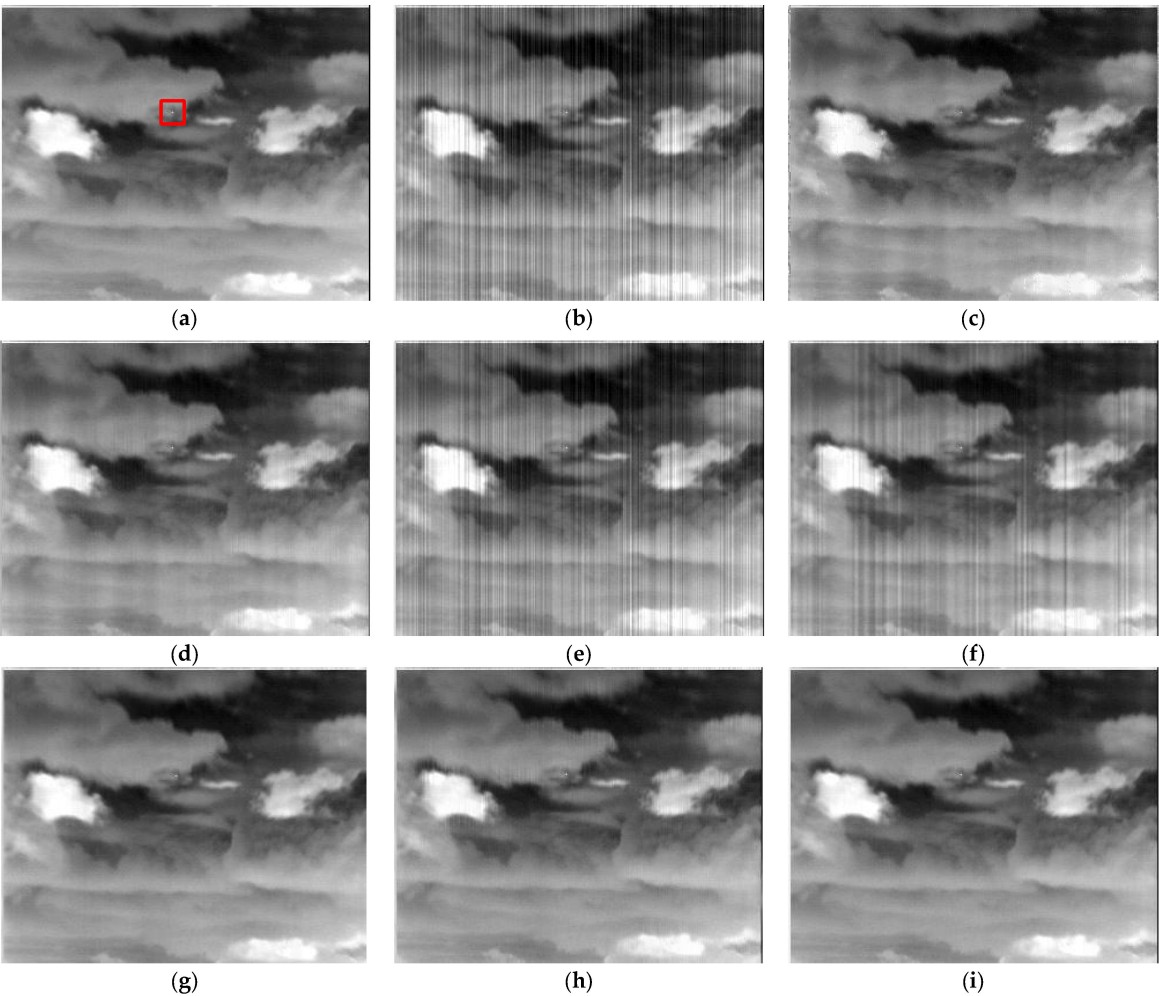

**Figure 14.** The nonuniformity correction results of Test-10: (**a**) the real, clean infrared dim small target image ($B_{\text{var}} = 2880$), the dim small target is located in the middle of the red box we manually marked; (**b**) infrared dim small target image with nonuniform noise added; (**c**) MHE; (**d**) 1D-GF; (**e**) SNRCNN; (**f**) ICSRN; (**g**) DLS-NUC; (**h**) SNRWDNN; (**i**) ours.

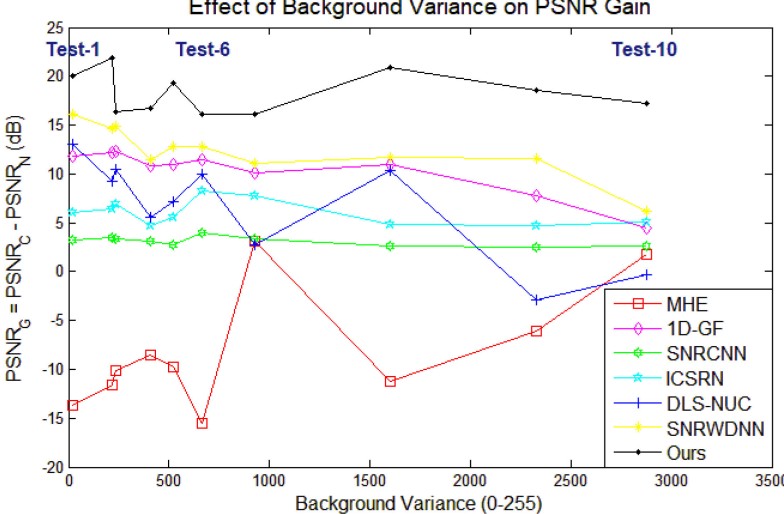

**Figure 15.** Line graph of each method performance on PSNR. The abscissa is the variance of the image background pixel value, and the ordinate is the difference between the corrected image PSNR and the nonuniformity image PSNR.

**Table 2.** Evaluation metric results of the methods on simulated data with different backgrounds.

| Experiment | Methods | RMSE | PSNR (dB) | SSIM | IR | SCR |
|---|---|---|---|---|---|---|
| | Noise Image | 14.73 | 24.7683 | 0.2990 | 0.1691 | 1.1610 |
| | MHE [14] | 71.15 | 11.0873 | 0.5375 | 0.0530 | 2.3561 |
| | 1D-GF [15] | 3.78 | 36.5751 | 0.9757 | 0.0137 | 4.2120 |
| Test − 1 ($B_{var} = 19$) | SNRCNN [27] | 10.09 | 28.0508 | 0.5531 | 0.0946 | 1.6311 |
| | ICSRN [28] | 7.30 | 30.8667 | 0.7791 | 0.0374 | 2.9876 |
| | DLS-NUC [29] | 3.26 | 37.8533 | 0.9756 | 0.0118 | 5.4496 |
| | SNRWDNN [30] | 2.32 | 40.8198 | 0.9909 | 0.0104 | 6.1065 |
| | Ours | 1.47 | 44.8026 | 0.9984 | 0.0099 | 6.7038 |
| | Noise Image | 12.63 | 26.1042 | 0.3894 | 0.2218 | 1.6516 |
| | MHE [14] | 75.02 | 10.6272 | 0.6675 | 0.0518 | 2.2701 |
| | 1D-GF [15] | 3.39 | 37.5167 | 0.9799 | 0.0318 | 3.2332 |
| Test − 6 ($B_{var} = 663$) | SNRCNN [27] | 8.04 | 30.0243 | 0.6737 | 0.1139 | 2.0920 |
| | ICSRN [28] | 4.91 | 34.3105 | 0.9033 | 0.0429 | 3.0031 |
| | DLS-NUC [29] | 4.01 | 36.0683 | 0.9741 | 0.0285 | 3.5229 |
| | SNRWDNN [30] | 2.90 | 38.8949 | 0.9817 | 0.0286 | 3.5026 |
| | Ours | 1.98 | 42.1847 | 0.9973 | 0.0271 | 3.7157 |
| | Noise Image | 15.39 | 24.3859 | 0.4616 | 0.1848 | 1.0346 |
| | MHE [14] | 12.54 | 26.1627 | 0.9194 | 0.0677 | 0.6328 |
| | 1D-GF [15] | 9.23 | 28.8261 | 0.9491 | 0.0600 | 0.8926 |
| Test − 10 ($B_{var} = 2880$) | SNRCNN [27] | 11.44 | 26.9595 | 0.6576 | 0.1227 | 1.1425 |
| | ICSRN [28] | 8.58 | 29.4648 | 0.8342 | 0.0731 | 1.2652 |
| | DLS-NUC [29] | 16.00 | 24.0461 | 0.7921 | 0.0521 | 1.2424 |
| | SNRWDNN [30] | 7.52 | 30.6026 | 0.9400 | 0.0605 | 0.8147 |
| | Ours | 2.12 | 41.6190 | 0.9958 | 0.0587 | 0.8182 |
| | Noise Image | 15.17 | 24.56 | 0.3562 | 0.1791 | 1.0014 |
| | MHE [14] | 45.90 | 16.39 | 0.7360 | 0.0622 | 2.0396 |
| | 1D-GF [15] | 4.89 | 34.82 | 0.9667 | 0.0311 | 2.3881 |
| Test-1–10 Average | SNRCNN [27] | 10.71 | 27.64 | 0.5824 | 0.1062 | 1.3462 |
| | ICSRN [28] | 7.75 | 30.59 | 0.7827 | 0.0515 | 2.0673 |
| | DLS-NUC [29] | 8.75 | 31.10 | 0.8967 | 0.0280 | 2.5928 |
| | SNRWDNN [30] | 3.89 | 36.87 | 0.9763 | 0.0293 | 2.6957 |
| | Ours | 1.87 | 42.90 | 0.9970 | 0.0275 | 2.8335 |
| | Noise Image | 1.6458 | 0.9419 | 0.0578 | 0.0728 | 0.4267 |
| | MHE [14] | 21.083 | 5.8496 | 0.1008 | 0.0285 | 0.9088 |
| | 1D-GF [15] | 1.8051 | 2.7607 | 0.0178 | 0.0237 | 1.2596 |
| Test-1–10 Standard Deviation | SNRCNN [27] | 1.6787 | 1.3723 | 0.0644 | 0.0423 | 0.5564 |
| | ICSRN [28] | 1.7758 | 2.0871 | 0.0887 | 0.0229 | 1.0465 |
| | DLS-NUC [29] | 6.3683 | 5.3226 | 0.0930 | 0.0226 | 1.4446 |
| | SNRWDNN [30] | 1.4669 | 2.9825 | 0.0141 | 0.0250 | 1.5914 |
| | Ours | 0.3615 | 1.9141 | 0.0012 | 0.0229 | 1.6913 |

At the same time, in order to verify the friendliness of our method to dim small targets, we used the clean infrared image SCR as the abscissa reference and created a line graph for each method performance on the SCR, as shown in Figure 17.

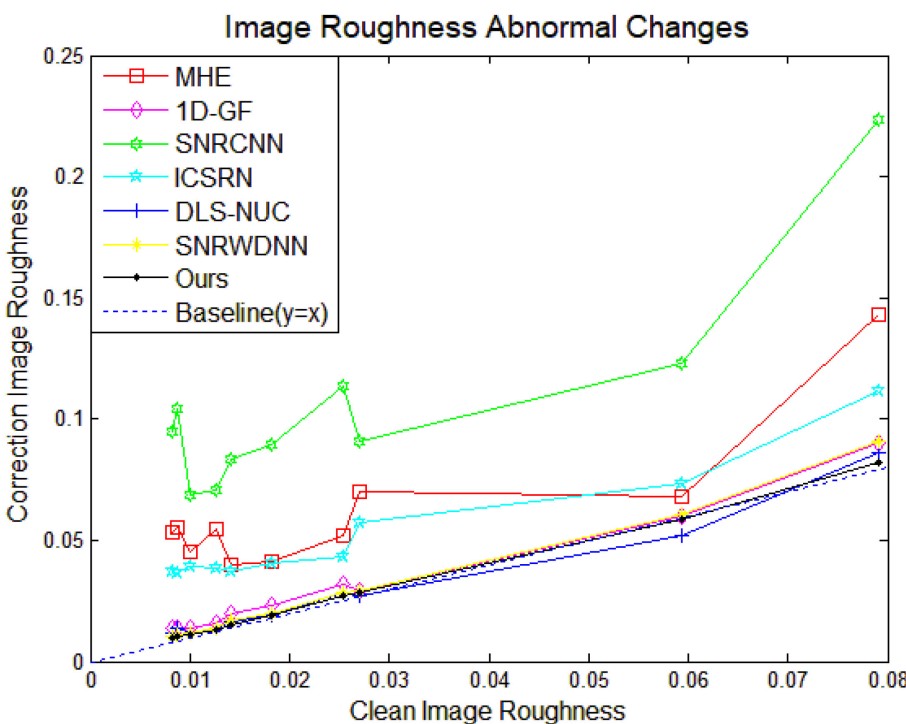

**Figure 16.** The line graph of each method performance on the IR. The abscissa is the clean image roughness, and the ordinate is the corrected image roughness. The better the fitting with the baseline, the lower the possibility of image over-smoothing.

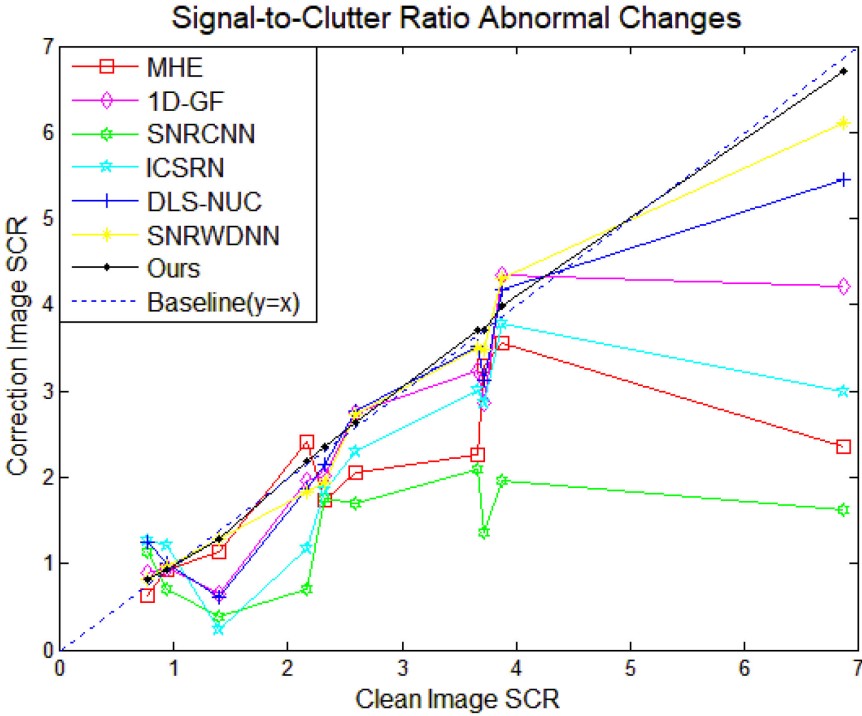

**Figure 17.** The line graph of each method performance on the SCR. The abscissa is the clean image SCR, and the ordinate is the corrected image SCR. The better the fitting with the baseline, the lower the possibility of removing dim small targets as noise points.

*3.5. Method Comparison on Real Data*

We compared the methods using a real nonuniformity infrared image, which was captured by an uncooled long-wave infrared camera and especially has many texture

details [29]. As shown in Figure 18a, we cropped out the (1:400, 1:480) area of the image as the real data 1, cropped out the (271:480, 481:640) area of the image as the real data 2, and the red dividing line in the figure is our cropping diagram. Figure 18b is the edge detection image, and the homogeneous region inside the blue box and the sharp region outside the blue box were used to calculate the evaluation metrics ICV and MRD.

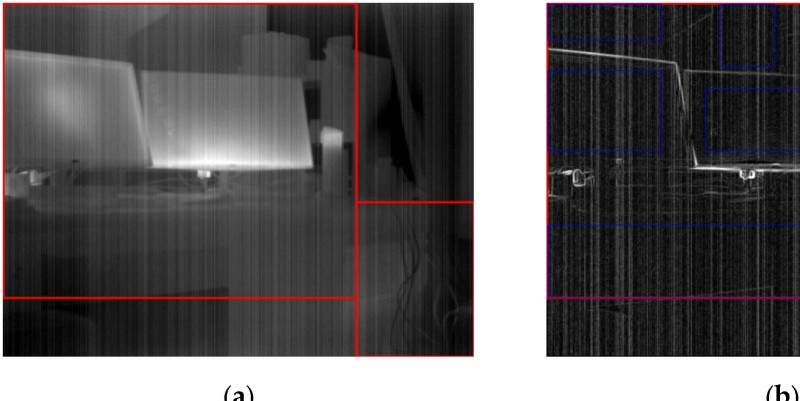

(a)    (b)

**Figure 18.** Real nonuniformity infrared image and its edge detection image: (**a**) real nonuniformity infrared image, upper left region as real data 1, bottom right region as real data 2; (**b**) the edge detection image.

Figure 19 shows the correction results of the real data 1 and the edge detection images of the correction results. Figure 20 shows the correction results of the real data 2 and the edge detection images of the correction results. In the experiment with real nonuniformity infrared images, we also compared the correction performance of each method according to the above evaluation metrics, as shown in Table 3.

**Table 3.** Evaluation metrics results of the methods on real data.

| Methods | Real Data 1 | | | Real Data 2 | | |
|---|---|---|---|---|---|---|
| | IR | ICV | MRD | IR | ICV | MRD |
| Noise Image | 0.1309 | 2.0711 | | 0.2907 | 2.4919 | |
| MHE [14] | 0.0859 | 1.9908 | 0.0647 | 0.1772 | 2.4026 | 1.9515 |
| 1D-GF [15] | 0.0786 | 2.0977 | 0.0647 | 0.1557 | 2.6051 | 0.2139 |
| SNRCNN [27] | 0.0803 | 2.0942 | 0.0501 | 0.1651 | 2.5885 | 0.1627 |
| ICSRN [28] | 0.0537 | 2.1001 | 0.0676 | 0.1097 | 2.588 | 0.2167 |
| DLS-NUC [29] | 0.0591 | 2.1453 | 0.0930 | 0.1142 | 2.6736 | 0.3349 |
| SNRWDNN [30] | 0.0790 | 2.0897 | 0.0648 | 0.1573 | 2.6013 | 0.2228 |
| Ours | 0.0792 | 2.1017 | 0.0649 | 0.1574 | 2.6247 | 0.2303 |

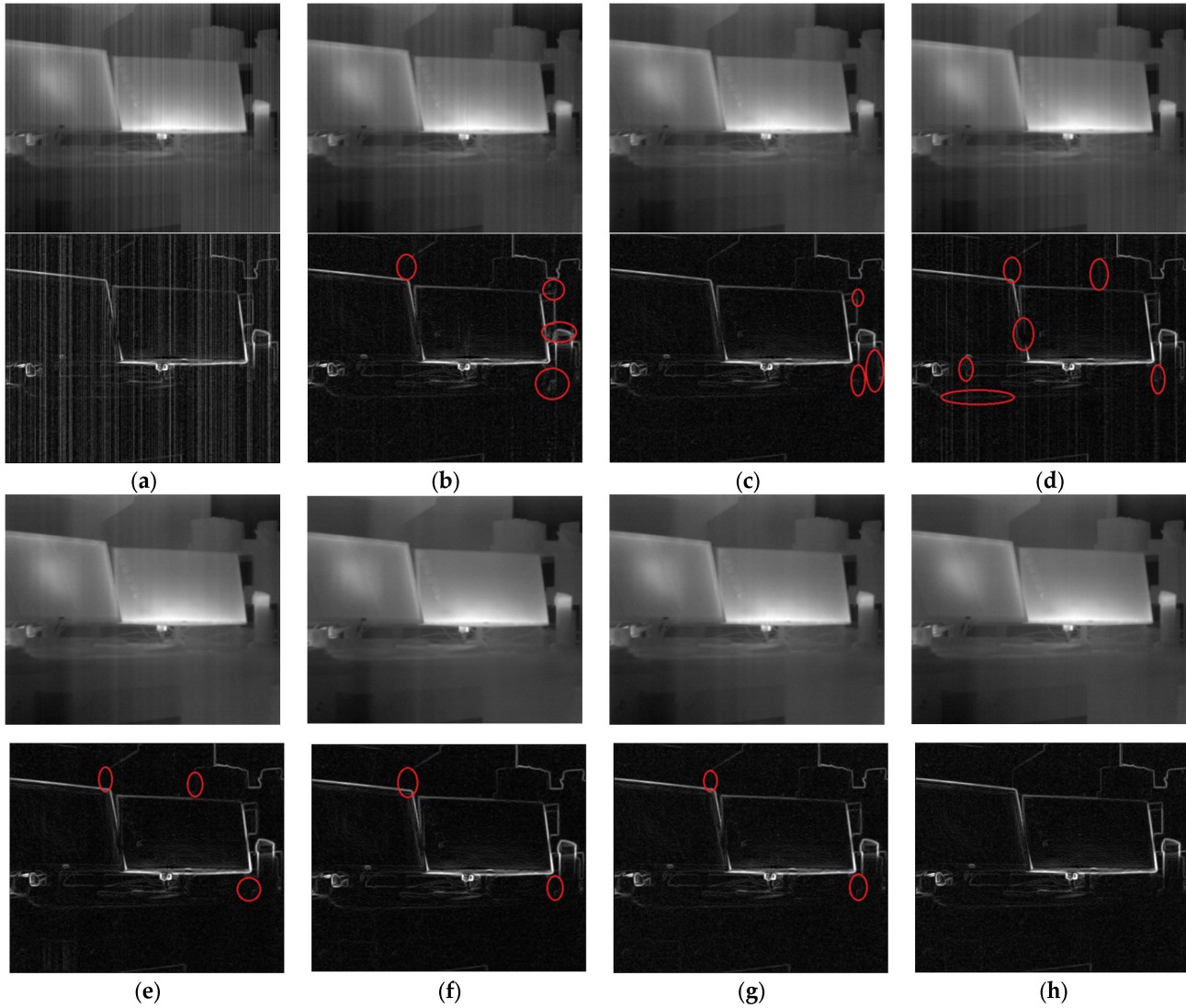

**Figure 19.** The correction results of the real data 1 and the edge detection images of the correction results. The red circle is our manual annotation of texture details in the image: (**a**) real data 1; (**b**) MHE; (**c**) 1D-GF; (**d**) SNRCNN; (**e**) ICSRN; (**f**) DLS-NUC; (**g**) SNRWDNN; (**h**) ours.

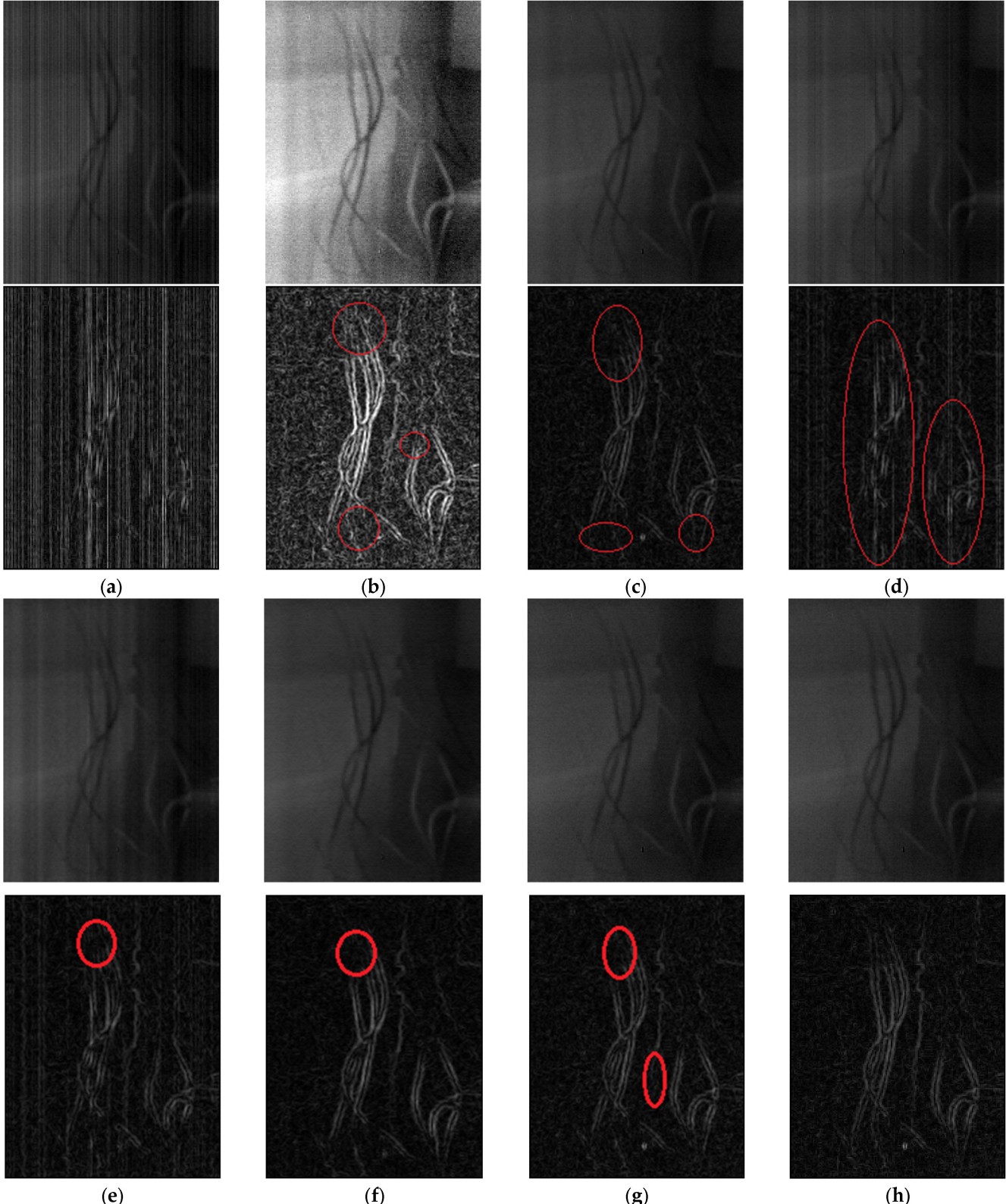

**Figure 20.** The correction results of the real data 2 and the edge detection images of the correction results. The red circle is our manual annotation of texture details in the image: (**a**) real data 2; (**b**) MHE; (**c**) 1D-GF; (**d**) SNRCNN; (**e**) ICSRN; (**f**) DLS-NUC; (**g**) SNRWDNN; (**h**) ours.

## 4. Discussion

The experimental results show that our method has excellent generalization to different backgrounds, can effectively remove nonuniform noise of different intensities, and was effectively verified on real data. In addition, our method can avoid the image over-smoothing problem, and mechanically ensures that texture details including weak objects will not be removed in the process of nonuniformity correction.

### 4.1. Analysis of Simulation Experiment Results for Different Intensities of Nonuniformity

In the simulation experiments for different intensities of nonuniformity, MHE reduces the IR, but it performs poorly in the PSNR and SSIM metrics. The 1D-GF method can perform effective correction on low- and medium-intensity nonuniformity images, but there is still a certain degree of residual strip noise. In addition, it is difficult for the 1D-GF to meet the needs of practical application with the image with high-intensity nonuniform noise. As a preliminary attempt of a deep learning method in the field of image nonuniformity correction, the SNRCNN method can remove nonuniform noise to a certain extent, but the correction ability cannot meet the needs of practical applications. Compared with SNRCNN, the correction ability of ICSRN is slightly improved, but it can only meet the needs of low-intensity nonuniformity correction. The correction ability of the DLS-NUC is comparable to the traditional classical algorithm 1D-GF, which can basically meet the nonuniformity correction requirements. The SNRWDNN has good performance on images with different intensities of nonuniformity.

Compared to the above methods, our method has the best correction performance for nonuniform noise of different intensities. The PSNR of our corrected image can be improved by at least 15 dB compared to the PSNR of the uncorrected image, and the SSIM of the simulation experiments all reached above 0.995. In addition, it can be seen from Figure 11 that the greater the intensity of nonuniformity, the higher the PSNR gain of MHE and DLS-NUC, while the lower the PSNR gain of the SNRCNN, ICSRN, and 1D-GF. At the same time, SNRWDNN and our method do not change much in PSNR gain. In comparison, our method has the best generalization performance to different intensities of nonuniform noise among these methods.

### 4.2. Analysis of Simulation Experiment Results for Different Backgrounds

In the experiment of Section 3.4, we added the same intensity of nonuniform noise to 10 infrared dim small targets images with different backgrounds. From Table 2 and Figure 15, it can be seen that our method not only has the highest average PSNR, but also has the highest PSNR of the correction results for each image. At the same time, our method also shows excellent performance compared to other methods on the SSIM. More notably, the standard deviation of the PSNR in our experiments is also the smallest, which shows that our method has an excellent generalization ability to infrared images with different backgrounds. In contrast, the standard deviation of the PSNR of MHE and DLS-NUC is high, indicating that their correction performance is very easily affected by different backgrounds. Furthermore, as can be seen from Figures 16 and 17, our method fits the baseline best in terms of IR and SCR, which demonstrates the effectiveness of our proposed mechanism based on noise parameter estimation. Compared with other methods for suppressing the image over-smoothing, our method fundamentally avoids the problem that dim small targets or texture details may be removed in the correction process through the innovation of the mechanism.

### 4.3. Analysis of Real Experimental Results

In the correction experiment for real nonuniformity infrared images, the MHE method, the GF method, and the SNRCNN method all have very obvious strip noise residuals. From the edge detection images of each correction result, other methods all have the phenomenon of losing texture details.

From the objective metrics in Table 3, compared with other methods, although our method performs poorly on IR, it is not far behind other methods. In the experiments of real data 1, DLS-NUC achieves the best performance on ICV, SNRWDNN achieves the best performance on MRD, and our method achieves a suboptimal performance on both metrics. At the same time, based on the correction results of real data 1 in Figure 19, it can be considered that our method achieved a very good balance on these two metrics. That is, our method can effectively remove strip noise and preserve image details well. In experiments on real data 2, we still achieve a suboptimal performance. However, on the MRD metric, both SNRCNN and ICSRN, which performed poorly in qualitative results, achieved top performance. We hypothesize that this is because the residual strip noise reduces the deviation between the correction result and the noisy image.

## 5. Conclusions

The main contribution of this paper is to propose a two-stage deep learning network based on a noise parameter estimation mechanism, which is able to perform nonuniformity correction on a single-frame infrared line-scan image. Our network model abandons the pixel-by-pixel estimation mechanism of the traditional end-to-end image reconstruction network, but estimates by column, so that when our network is applied to dim small target images, it can ensure that dim small targets are not removed as noise points. According to the infrared line-scan imaging mechanism, we produced a noise parameter estimation and image reconstruction dataset and trained our nonuniformity correction model with this dataset. In general, compared with existing methods, our method has many advantages, including excellent performance, intelligence, and friendliness to texture details and dim small targets. Of course, our model still has some room for improvement in real time. Therefore, in the future work, we plan to simplify our model by migrating learning to improve the processing speed of the model and obtain better practical value.

**Author Contributions:** Conceptualization, T.W.; methodology, T.W.; software, T.W.; validation, T.W. and Q.Y.; formal analysis, T.W.; investigation, T.W.; resources, F.C. and M.L.; data curation, T.W.; writing—original draft preparation, T.W.; writing—review and editing, T.W., Q.Y. and F.C.; visualization, T.W.; supervision, M.L.; project administration, Z.L. and W.A.; All authors have read and agreed to the published version of the manuscript.

**Funding:** This research was funded by the National Natural Science Foundation of China, grant number 62001478.

**Institutional Review Board Statement:** Not applicable.

**Informed Consent Statement:** Not applicable.

**Data Availability Statement:** The synthetic data underlying this article will be shared upon reasonable request to the corresponding author.

**Conflicts of Interest:** The authors declare no conflict of interest.

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
