# Peer review of "Noise Parameter Estimation Two-Stage Network for Single Infrared Dim Small Target Image Destriping"

_remotesensing, doi:10.3390/rs14195056_

Round 1

Reviewer 1 Report

In this paper, the authors propose a new method for infrared line-scan dim small target image nonuniformity correction. This method is based on the use of a two-stage fully convolutional network. The applied method as well as the two-stage fully convolutional network architecture are explained in a clear and consistent way. A comparison of the proposed method with the most commonly used single-frame infrared image nonuniformity correction methods on simulated nonuniformity infrared dim small target images and real nonuniformity infrared images was performed. The main drawback of this paper is the following:

-          When comparing the proposed method with other methods on real data, the image roughness (IR) metric of the obtained results was used. The results obtained using the proposed method are slightly worse than the results obtained using the CRL (Cascade Residual Learning) method and the GF (Guided Filter) method. Regardless of the slightly worse results, the authors justify the use of the proposed method by showing that it does not have the image over-smoothing problem of other methods, and completely retains the texture details. Although the mentioned advantage of the proposed method can be visually seen in the attached images, it would still be desirable to quantify it through some additional selected metric that would complement the results obtained by applying the IR metric (Table 2). This would clearly favor the use of the proposed method over the CRL and GF methods.

It should also be noted that the paper did not state that the authors used batch normalization when training the two-stage fully convolutional network. In case the authors have not used this type of normalization, they are recommended to apply it, as it can improve the results.

Author Response

Point 1: Although the mentioned advantage of the proposed method can be visually seen in the attached images, it would still be desirable to quantify it through some additional selected metric that would complement the results obtained by applying the IR metric (Table 2)..

Response 1: Thank you for the suggestion. We added the inverse coefficient of variation (ICV) metric and the mean relative deviation (MRD) metric to analyze the experimental results on real data. ICV is a metric for evaluating the smoothness of an image homogeneous region, and MRD is a metric for evaluating the relative distortion of sharp region. Detailed definitions of these two metrics are located on page 10 of the revised version. And the metric evaluation results are located in Table 3 on page 22 ( We have added a new set of simulation experiments, so the corresponding results are in Table 3 of the revised version). Although ICV and MRD also have certain limitations as no reference metrics, their adoption is very helpful to support our conclusions. Thank you again for the suggestion.

Point 2: 2.It should also be noted that the paper did not state that the authors used batch normalization when training the two-stage fully convolutional network. In case the authors have not used this type of normalization, they are recommended to apply it, as it can improve the results.

Response 2: Thank you for your advice and guidance. It is true that batch normalization in the network tends to improve the results. But when we adopted batch normalization in our two-stage network, the fitting performance dropped. We speculate that this may be due to the fact that although our task needs to utilize feature information, it is more biased towards nonlinear estimation rather than feature extraction.

Reviewer 2 Report

Compared with other methods, the method in this paper has a certain improvement and can solve the problem of image correction. However, there are still some issues that need to be explained.

1. The citations in this paper are old, which does not indicate the novelty of the research in this paper.

2. The lack of recent research and analysis in the Introduction does not account for the superiority of this method.

3.The font in Figure 6 is too large.

4. It is recommended to mark the source of the comparative experiment in the table.

5.Why does the network model in this article choose convolutional neural networks instead of other networks?

Author Response

Point 1: The citations in this paper are old, which does not indicate the novelty of the research in this paper. 

Response 1: Thank you for this valuable feedback. We again checked some of the latest relevant research literature and found many interesting approaches. However, most are not open source yet. But we still found several relatively new open source methods and compared them with ours. The new experimental results have been placed in our revised version. Thanks again for your valuable feedback.

Point 2: The lack of recent research and analysis in the Introduction does not account for the superiority of this method.

Response 2: Thank you very much for pointing this out, it helped us a lot. In the revised introduction, we have added some new research methods. At the same time, we also conduct a more scientific classification and review of the existing methods. On this basis, we also more condensed the contribution of our method. Thanks again for the valuable feedback.

Point 3: The font in Figure 6 is too large..

Response 3: We apologize for the oversight on Figure 6 and appreciate your pointing out. We have reproduced Figure 6, which is located on page 9 of the revised version.

Point 4: It is recommended to mark the source of the comparative experiment in the table.

Response 4: Thanks a lot for the suggestions, we have cited the source of each method in the table.

Point 5: Why does the network model in this article choose convolutional neural networks instead of other networks?

Response 5: On the one hand, this is mainly because the effectiveness of convolutional neural networks in various fields has been verified. On the other hand, in principle, our task needs to extract useful information from images for nonlinear estimation of noise parameters, and convolutional neural networks perform very well in both feature extraction and nonlinear estimation. Of course, we can also try to use other popular networks such as Transformer to achieve our task. However, although Transformer has a lot of excellent performance, especially in solving long-term dependencies. Compared with RNN and CNN, it loses the ability to capture local features, which may not be very friendly for our task.

Reviewer 3 Report

This paper proposes a two-stage learning network based on the imaging mechanism of infrared line-scan system, it is useful for image nonuniformity correction. Some comments are as follows:

1.    In the simulations, some images with complex background should be analyzed.

2.    In the experiments, some images containing small target should be also analyzed.

3.    In the experiments, the edge detection images are actually the gradient maps? The dynamic ranges of these gradient maps are consistent or not?

Author Response

Point 1: In the simulations, some images with complex background should be analyzed.

Response 1: Thank you very much for the suggestion, it helped us a lot. We add a new set of simulation experiments to evaluate the performance of each method on different background complexities. The results of this set of experiments are on pages 14-19 of the revised version of the paper. Experimental results show that our method has excellent generalization performance to different backgrounds.

Point 2: In the experiments, some images containing small target should be also analyzed.

Response 2: Thank you very much for the suggestion. Since there is no public nonuniformity infrared dim small target images, we can only perform experimental analysis by adding nonuniformity noise to the clean infrared dim small target images.

Point 3: In the experiments, the edge detection images are actually the gradient maps? The dynamic ranges of these gradient maps are consistent or not?.

Response 3: Yes, the edge detection images are actually the gradient maps, and the dynamic ranges of these gradient maps are all 0-255.

Reviewer 4 Report

This method proposed two-staged learning network on the basis of imaging mechanism of infrared line-scan system. Authors used multi-scale features extraction unit and designed a gain correction sub-network and an offset correction sub-network. They cascaded two sub-networks into two-stage network and trained it. The given presentation is good, however, I have few concerns that are given below:

The introduction is written very weak and require intense modification in terms of some visual representation and their explanation to highlight the generic flow of the proposed method.

There are no challenges explained to highlight and provide the solution to address those challenges.

There is no visual explaining to show the working flow of the proposed two staged network. It is recommended to provide a detailed framework showing the actual steps.

The given captions to figures and tables are not meaningful and need to be elongated for explanation.

Some recent deep learning-based literature can be included such as https://doi.org/10.1155/2022/2993184, 10.1109/ACCESS.2018.2874767,  and get expression of how made DL applicable in your problem.

I recommend the authors to proofread the paper with English team as I examined the language weakness of writing.

I observed that authors used pre-trained networks and there is no major modification or change in the network for the given problem. Authors should highlight the actual contributions and novelty.

Author Response

Point 1: The introduction is written very weak and require intense modification in terms of some visual representation and their explanation to highlight the generic flow of the proposed method. 

Response 1: Thank you very much for the suggestions. In the revised introduction, we have added some new research methods. At the same time, we also conduct a more scientific classification and review of the existing methods. In terms of some visual representation and their explanation, we also made many modifications.

Point 2: There are no challenges explained to highlight and provide the solution to address those challenges.

Response 2: Thank you very much for the suggestions. We reorganize the research background in the introduction and explain the challenges in the field of infrared dim small target image research in detail.

Point 3: There is no visual explaining to show the working flow of the proposed two staged network. It is recommended to provide a detailed framework showing the actual steps. 

Response 3: Thank you very much for the suggestions. We reworked our network structure diagram as shown in Figure 5, page 7 of the revised version.

Point 4: The given captions to figures and tables are not meaningful and need to be elongated for explanation.

Response 4: Thank you very much for the suggestions, it helped us a lot. Based on your suggestion, we have revised the captions of the figures and tables.

Point 5: Some recent deep learning-based literature can be included such as https://doi.org/10.1155/2022/2993184, 10.1109/ACCESS.2018.2874767,  and get expression of how made DL applicable in your problem. 

Response 5: Thank you very much for the suggestions. Based on your suggestion, we read and learned the deep learning design ideas in the article, and cite it in the revised edition to better support the necessity of our method.

Point 6: I recommend the authors to proofread the paper with English team as I examined the language weakness of writing.

Response 6: We apologize for the poor language of our manuscript. We worked on the manuscript for a long time and the repeated addition and removal of sentences and sections obviously led to poor readability. We have now worked on both language and readability and have also involved English teachers for language corrections. We really hope that the flow and language level have been substantially improved.

Point 7: I observed that authors used pre-trained networks and there is no major modification or change in the network for the given problem. Authors should highlight the actual contributions and novelty. 

Response 7: Thank you very much for the suggestions. Our method is a two-stage network, the two sub-networks are pre-trained to better fit our task. And the two pretrained sub-networks are designed and pretrained by ourselves. Moreover, based on your suggestion, we have further refined our contributions, which is located on the end of page 2 to the beginning of page 3.

Round 2

Reviewer 2 Report

The author has been revised and improved according to the review comments. 

Reviewer 3 Report

The authors have answered my concerns,and the paper can be accepted.